# The molecular basis of JAK/STAT inhibition by SOCS1

Nicholas P.D. Liau[1,2], Artem Laktyushin[1,2], Isabelle S. Lucet[1,2], James M. Murphy[1,2], Shenggen Yao[2], Eden Whitlock[1,2], Kimberley Callaghan[1,2], Nicos A. Nicola[1,2], Nadia J. Kershaw[1,2] & Jeffrey J. Babon [1,2]

The SOCS family of proteins are negative-feedback inhibitors of signalling induced by cytokines that act via the JAK/STAT pathway. SOCS proteins can act as ubiquitin ligases by recruiting Cullin5 to ubiquitinate signalling components; however, SOCS1, the most potent member of the family, can also inhibit JAK directly. Here we determine the structural basis of both these modes of inhibition. Due to alterations within the SOCS box domain, SOCS1 has a compromised ability to recruit Cullin5; however, it is a direct, potent and selective inhibitor of JAK catalytic activity. The kinase inhibitory region of SOCS1 targets the substrate binding groove of JAK with high specificity and thereby blocks any subsequent phosphorylation. SOCS1 is a potent inhibitor of the interferon gamma (IFNγ) pathway, however, it does not bind the IFNγ receptor, making its mode-of-action distinct from SOCS3. These findings reveal the mechanism used by SOCS1 to inhibit signalling by inflammatory cytokines.

---

[1] Walter and Eliza Hall Institute, 1G Royal Parade, Parkville, VIC 3052, Australia. [2] The University of Melbourne, Royal Parade, Parkville, VIC 3050, Australia. These authors contributed equally: Nadia J. Kershaw, Jeffrey J. Babon. Correspondence and requests for materials should be addressed to N.J.K. (email: kershaw@wehi.edu.au) or to J.J.B. (email: babon@wehi.edu.au)

The immune response is largely controlled via the action of specific cytokines. Exposure to cytokines initiates an intracellular signalling cascade driven by activation of a family of receptor-bound tyrosine kinases known as the JAKs (Janus Kinases), which directly activate a family of transcription factors, the STATs (Signal Transducers and Activators of Transcription). STATs then drive the transcription of cytokine-inducible genes. Prolonged signalling, particularly by inflammatory cytokines, is detrimental to both the cell and the organism, and is therefore tightly regulated, in particular, by the SOCS (Suppressors of Cytokine Signalling) family of proteins. There are eight SOCS proteins encoded in the human genome, SOCS1-7 and CIS[1]. All eight are defined by the presence of an SH2 domain and a short, C-terminal domain, the SOCS box[1] (Fig. 1a). The SOCS box of all SOCS proteins are found associated with an adapter complex, elonginBC[2]. This association allows recruitment of an E3 ubiquitin ligase scaffold (Cullin5) to catalyse the ubiquitination of signalling intermediates recruited by their SH2 domains[3]. In addition to their ubiquitin ligase activity, SOCS1 and SOCS3 are unique in also having the ability to directly inhibit the kinase activity of JAK[4]. This activity relies upon a short motif, which is immediately upstream of the SH2 domain, known as the KIR (kinase inhibitory region).

SOCS1 is the most potent member of the SOCS family[5–7] and is the primary regulator of a number of cytokines involved in the immune response, in particular IFNγ. Genetic deletion of SOCS1 is lethal. SOCS1$^{-/-}$ mice die at 2–3 weeks of age from general inflammation and necrosis of the liver[8]. Neonatal death can be rescued by genetic deletion of IFNγ[9], although SOCS$^{-/-}$ IFNγ$^{-/-}$ mice still develop fatal inflammatory disease, polycystic kidneys and die earlier than WT littermates[10]. Early neonatal death can also be rescued by genetic deletion of STAT1, STAT4 or STAT6 although such mice still succumb to lethal inflammatory disease within 1–3 months. Studies such as these have identified SOCS1 as a potent inhibitor of IFNα/β/γ[9,11], IL-12/23[12], IL4/13[13] and cytokines that signal through the γ$_c$ chain (IL-2 family cytokines)[14].

SOCS1 is a tumour suppressor[15] and therefore downregulation of SOCS1 can play a role in the progression of human cancer. It is found silenced in primary tumours in >50% of hepatocellular carcinoma[16], 44% of gastric carcinoma[17], 75% of melanoma[18] and 40% of hepatoblastoma cases. In addition to solid tumours, SOCS1 is also silenced in 60% of acute myeloid lymphoma[19], 62% of multiple myeloma[20] and >50% of chronic myeloid leukemia cases. SOCS1 may also play a role in the pathogenesis of myeloproliferative disease, being silenced in 14–25% of essential thrombocythemia, 11–13% of polycythemia vera and 17% of primary myelofibrosis cases[21].

Here we have performed a structural and biochemical analysis of SOCS1 activity. We have determined its mode-of-action by solving two X-ray crystal structures of SOCS1 bound to its physiological targets (both the ElonginB/C adapter complex and also the JAK1 kinase domain). SOCS1 is shown to be a direct inhibitor of the catalytic activity of JAK1, JAK2 and TYK2 but not JAK3. Inhibition by SOCS1 is an order-of-magnitude more potent than by SOCS3. Our structure reveals that SOCS1 inhibits these three

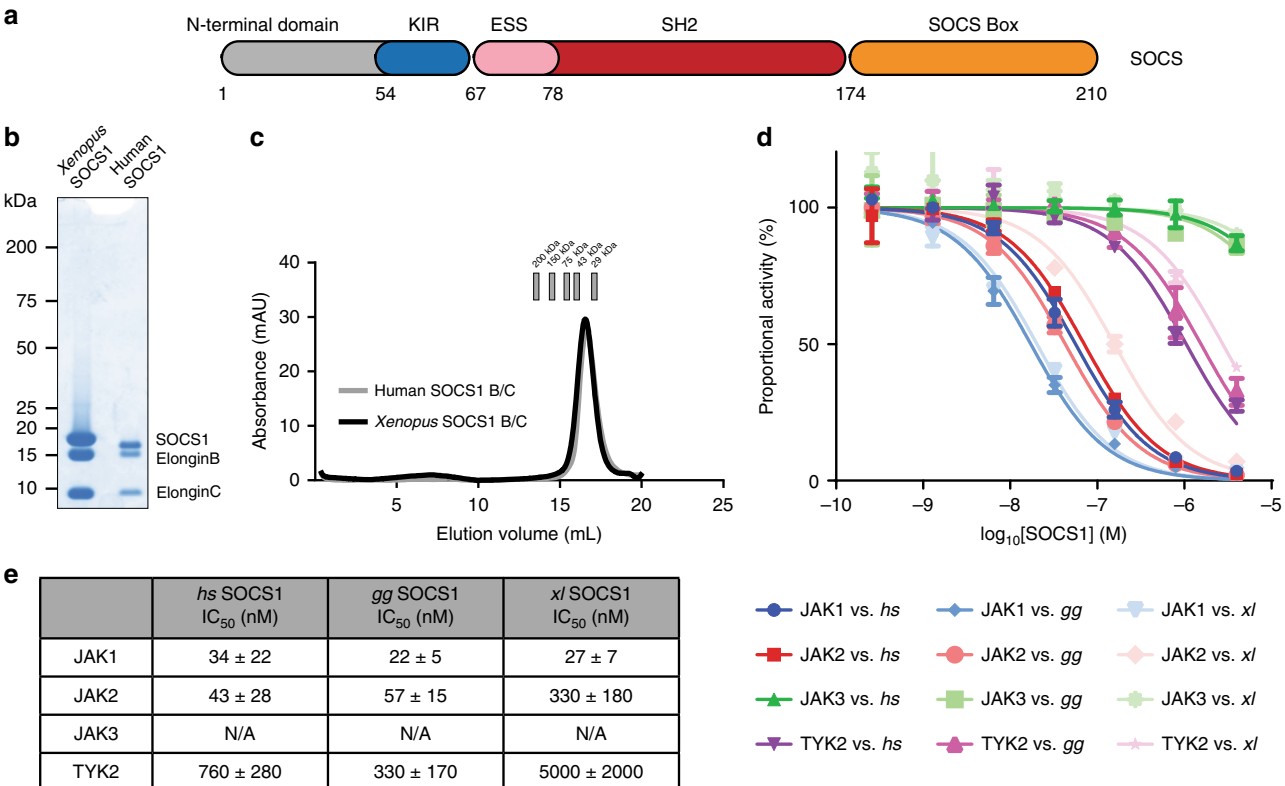

| | *hs* SOCS1 IC$_{50}$ (nM) | *gg* SOCS1 IC$_{50}$ (nM) | *xl* SOCS1 IC$_{50}$ (nM) |
|---|---|---|---|
| JAK1 | 34 ± 22 | 22 ± 5 | 27 ± 7 |
| JAK2 | 43 ± 28 | 57 ± 15 | 330 ± 180 |
| JAK3 | N/A | N/A | N/A |
| TYK2 | 760 ± 280 | 330 ± 170 | 5000 ± 2000 |

**Fig. 1** SOCS1 is a direct inhibitor of JAK kinase activity. **a** Schematic representation of the SOCS1 domain architecture. SOCS1 consists of an unstructured N-terminal region, followed by the kinase inhibitory region (KIR), an extended SH2 subdomain (ESS), a SH2 domain and a SOCS box domain. **b** Purified recombinant SOCS1 in complex with Elongins B and C shown on Coomassie stained SDS-PAGE gel. **c** Gel filtration analysis of purified recombinant SOCS1 in complex with Elongins B and C. **d** Kinase inhibition assays indicate that SOCS1 from *Homo sapiens* (*hs*), *Gallus gallus* (*gg*) and *Xenopus laevis* (*xl*) inhibits the kinase domains of JAK1, JAK2 and TYK2, but not JAK3. **e** IC$_{50}$ values of SOCS1 and SOCS3 constructs against JAKs as determined by kinase inhibition assays. The error bars shown in **d** represent the range of the data from two technical replicates whilst errors given in **e** are the standard error of the mean from 3 independent experiments

**Table 1 Data collection and refinement statistics (molecular replacement)**

| | SOCS1/JAK1 (6C7Y) | SOCS1/ElonginBC (6C5X) |
|---|---|---|
| Data collection | | |
| Space group | P4$_1$2$_1$2 | P2$_1$2$_1$2$_1$ |
| Cell dimensions | | |
| $a, b, c$ (Å) | 83.93, 83.93, 161.44 | 61.12, 79.99, 132.75 |
| $\alpha, \beta, \gamma$ (°) | 90, 90, 90 | 90, 90, 90 |
| Resolution (Å) | 40–2.50 (2.59–2.50)[a] | 49–3.11 (3.22–3.11)[a] |
| $R_{sym}$ or $R_{merge}$ | 0.1131 (1.663) | 0.2006 (1.071) |
| $I/\sigma I$ | 16.09 (1.67) | 6.88 (1.54) |
| Completeness (%) | 99.89 (99.80) | 99.1 (94.0) |
| Redundancy | 8.3 (8.4) | 4.8 (4.8) |
| Refinement | | |
| Resolution (Å) | 40–2.50 (2.59–2.50) | 49–3.11 (3.22–3.11) |
| No. reflections | 171,587 (17,042) | 58,956 (5499) |
| $R_{work}/R_{free}$ | 0.206/0.241 | 0.236/0.271 |
| No. atoms | | |
| Protein | 3075 | 5068 |
| Ligand/ion | 45 | — |
| Water | 35 | 1 |
| B-factors | | |
| Protein | 85.22 | 61.30 |
| Ligand/ion | 79.27 | — |
| Water | 56.67 | 41.50 |
| R.m.s. deviations | | |
| Bond lengths (Å) | 0.002 | 0.004 |
| Bond angles (°) | 0.55 | 0.98 |

Each structure was determined from one crystal
[a]Values in parentheses are for highest resolution

JAK family members by blocking the substrate binding groove on JAK, acting as a pseudosubstrate. The KIR of SOCS1 is a highly evolved inhibitor of JAK and mutation of any residue within this motif, including the histidine residue that mimics the substrate tyrosine, leads to a significant decrease in affinity. SOCS1 does not require cytokine-receptor recruitment to inhibit signalling by its primary target, IFNγ. Finally, we show that SOCS1 can target unphosphorylated (and therefore inactive) JAK, suggesting a model in which SOCS1 inhibits signalling by preventing JAK autophosphorylation in addition to downstream substrate phosphorylation.

## Results

**SOCS1 is a potent inhibitor of JAK1 and JAK2.** SOCS1 is arguably the most important of the eight SOCS family members, but has hitherto resisted attempts to be recombinantly expressed and purified as a soluble, active protein. To overcome this, we tested a large array of different SOCS1 constructs, expression and purification conditions. Successful expression and purification was achieved by deleting the first 52 residues, co-expressing SOCS1 with its native ligands, Elongins B and C and including a phosphotyrosine mimetic during purification (Fig. 1a). In this way, stable and homogenous SOCS1/BC complexes were produced using human, chicken (*Gallus gallus*) and frog (*Xenopus laevis*) SOCS1 (Fig. 1b, c).

To test whether SOCS1 can directly inhibit JAK catalytic activity, a series of in vitro kinase assays were performed, measuring the tyrosine phosphorylation of a peptide substrate by purified JAK kinase domains and the inhibition of such

phosphorylation by recombinant SOCS1[22,23]. SOCS1 from all three species could directly inhibit the catalytic activity of JAK1, JAK2 and TYK2 but not JAK3 (Fig. 1d, e). Our data revealed that SOCS1 is a particularly potent inhibitor of JAK1 and JAK2 (IC$_{50}$ = 30, 40 nM, respectively), an order of magnitude more potent than the related family member SOCS3[24] (Fig. 1d, e).

**SOCS1 inhibits JAK by blocking substrate binding.** To structurally characterise the mechanism of SOCS1 inhibition of JAK kinase activity, we solved the structure of *Gallus gallus* SOCS1, without its SOCS Box domain, in complex with the human JAK1 kinase domain bound to ADP, using X-ray crystallography (*gg*SOCS1$^{\Delta SOCSbox}$-JAK1$^{KD}$). Attempts to crystallise a complex containing human SOCS1 were unsuccessful, however, the *Gallus gallus* and human orthologues share 72% identity and 89% similarity and display similar efficacy in JAK inhibition (Fig. 1d, e), indicating that it is a good model for the human protein. Crystals of a 1:1 complex were obtained and the data processed to 2.5 Å (Table 1). Phases were obtained by molecular replacement using the JAK1 kinase domain (PDB: 3EYH) and *Xenopus laevis* SOCS1 (PDB: 6C5X, this manuscript).

As shown in Fig. 2a, SOCS1 binds to JAK using both its SH2 domain and kinase inhibitory region. The interface between the two proteins is 1159 Å$^2$ and the KIR of SOCS1 is responsible for approximately half of this buried surface (Fig. 2b). In addition to the KIR, a surface on the distal face of the SOCS1 SH2 domain (relative to the pTyr binding site), which corresponds primarily to the BC loop[24], is responsible for the remaining buried surface, contacting the JAK1 GQM motif[22]. Overall, the complex adopts a very similar conformation to that of the previously solved JAK2-SOCS3-gp130 complex (PDB: 4GL9)[22], overlaying with an RMSD of 1.1 Å over 276 residues (JAK) and 1.6 Å over 117 residues (SOCS) (Supplementary Figure 1A). Both the KIR and SH2 domain-driven interactions are conserved in the SOCS3/JAK2 structure with similar overall geometry.

The only significant conformational change in JAK1 upon SOCS1-binding is seen in the glycine-rich loop. The position of this loop is somewhat variable (as seen by comparing JAK structures) but adopts a more closed position when SOCS1 is bound. (Fig. 2c). SOCS1 displays only minor structural changes upon JAK1 binding (Fig. 2d). The largest difference is that the KIR becomes ordered in the JAK-bound structure whereas there is no associated electron density for it when in the apo-form.

As shown in Fig. 2e, most of the individual contacts between the KIR and JAK are conserved in both SOCS1 and SOCS3. The significantly improved resolution of the SOCS1/JAK1 structure allows a detailed atomic-level analysis of this interaction (Supplementary Figure 1B) and shows that it is mediated by a continuous six residue segment of the KIR (His54 to Arg59), corresponding to residues −1 to +5 of this motif based on earlier definitions[4]. Each one of these six amino-acids is involved in at least one significant polar and/or hydrophobic interaction (Fig. 3a). The first of these, His54, is sandwiched between two planar sidechains from JAK1, the upper sidechain is from His885 in the glycine-rich loop of the kinase and the lower sidechain is Pro1044. Pro1044 is highly conserved in all tyrosine kinases and provides the platform upon which a substrate tyrosine sits during catalysis. Indeed, His54 occupies the substrate-binding site[25], mimicking a substrate tyrosine and indicating that SOCS1 inhibits JAK1 by blocking peptide substrate (but not ATP) access (Fig. 3b). In SOCS3, when this residue is mutated to tyrosine it is efficiently phosphorylated by JAK and the same is true for SOCS1 (Supplementary Figure 1C). Phe55 is the P + 1 mimicking residue and, together with Phe58, form the bulk of the hydrophobic interactions (with Phe1046, Thr1100 and Val1101 from JAK1)

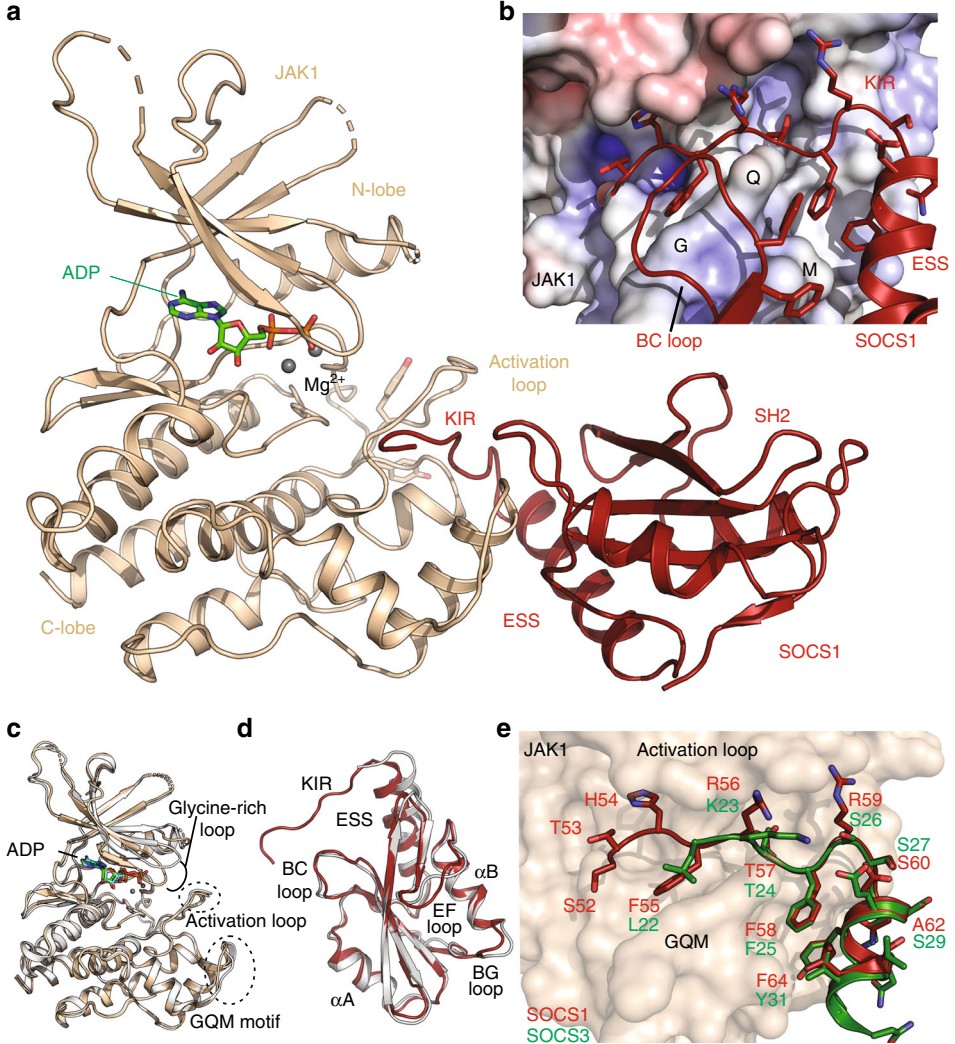

**Fig. 2** The structure of a SOCS1/JAK1/ADP complex. **a** Cartoon representation of the JAK1 (beige) /SOCS1 (red) /ADP (green) complex. The SOCS1 kinase inhibitory region (KIR), the extended SH2 subdomain (ESS) as well as the SH2 domain proper (but not its phosphotyrosine-binding groove) all make substantial contacts with the C-lobe of the JAK1 kinase domain. Notably, JAK1 Tyr 1034 and 1035 in the activation loop are unphosphorylated in this structure. **b** Details of key SOCS1 interactions centred around the JAK1 GQM motif (electrostatic surface view). **c** JAK1 from the JAK1/SOCS1/ADP complex structure (beige) is shown overlaid with a JAK1/ADP structure (white, PDB: 5KHW). Small perturbations in the JAK1 activation loop and GQM motif can be seen between the two structures as well as a larger perturbation in the glycine-rich loop. The JAK1 activation loop is seen in the active conformation both in the presence and absence of SOCS1, despite being unphosphorylated in the SOCS1 bound structure. **d** Ribbon diagram of the structure of SOCS1 in the presence (red) and absence (white) of JAK1. The major conformational change to SOCS1 upon JAK1 binding is an ordering of the kinase inhibitory region. **e** The SOCS1 KIR is shown overlaid with the SOCS3 KIR (green) (PDB ID: 4GL9). Residue numbering of SOCS1 orthologues in all figures refers to the analogous residue in the human sequence

that wedge the KIR between the activation loop and the αG helix as well as providing three mainchain-derived hydrogen bonds. Phe58 sits in a hydrophobic pocket formed by the interface of JAK1 and SOCS1. Additional contacts are formed via hydrogen bonds between the sidechains of Arg56 and Arg59 with residues in the activation loop as well as a hydrogen bond between the side chains of Thr57 (SOCS1) and Ser1056 (JAK1). In order to investigate the relative contribution of specific residues within the KIR, mutagenesis was performed. As shown in Fig. 3d, deletion of the KIR, or mutation of Phe58, lead to complete loss of inhibitory activity. Mutation of Phe55 leads to a >50-fold increase in $IC_{50}$, while the other mutations had a significant (approximately 10-fold) effect. These data indicate that each residue in the KIR is important for the interaction with JAK, a conclusion supported by the high sequence conservation seen in the KIR of all vertebrate orthologues (Fig. 3e).

**The GQM motif is a SOCS1/SOCS3 binding epitope.** The majority of the surface on JAK1 that interacts with SOCS1 is comprised of the activation loop and the αG helix (Fig. 3a). The αG helix includes the GQM motif (Gly1097-Met1099), previously shown to be required for SOCS3 binding[24]. This motif is conserved between JAK1, JAK2 and TYK2, but is absent in JAK3 in all vertebrates. The GQM motif and nearby residues form a network of interactions with SOCS1 involving both the kinase inhibitory region and the SH2 domain (Fig. 3c), particularly the BC loop (the canonical phosphotyrosine binding loop within the SOCS1 SH2 domain). It is the opposing face of this loop that contacts JAK1 at the GQM sequence. The central glutamine of the GQM motif (Gln1098) forms two hydrogen bonds with mainchain carbonyl groups of Gln108 and Cys111 and its sidechain is stacked onto the aromatic ring of Phe112, while the sidechain of Met1099 forms a hydrophobic contact with Asp105

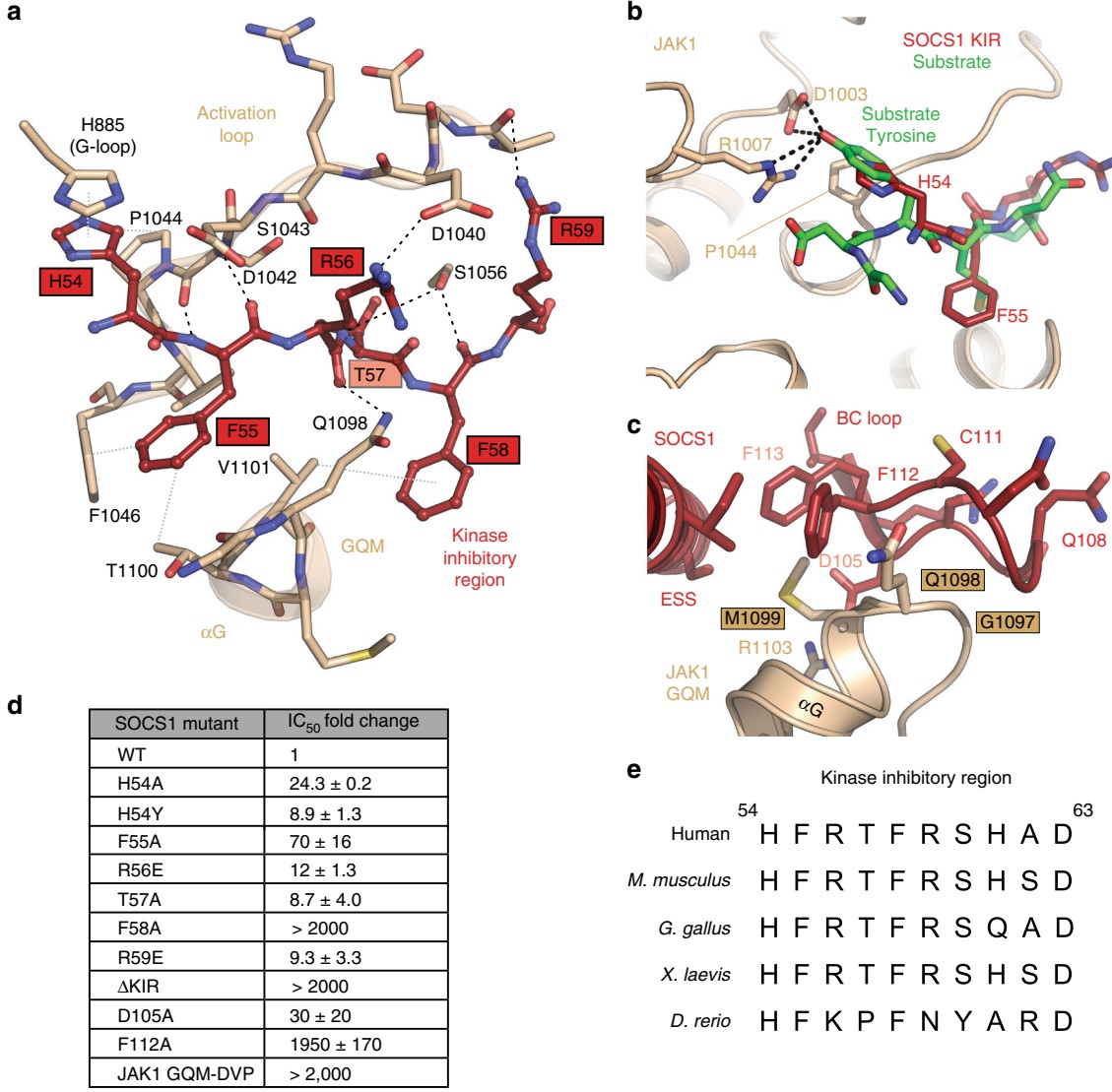

| SOCS1 mutant | IC$_{50}$ fold change |
|---|---|
| WT | 1 |
| H54A | 24.3 ± 0.2 |
| H54Y | 8.9 ± 1.3 |
| F55A | 70 ± 16 |
| R56E | 12 ± 1.3 |
| T57A | 8.7 ± 4.0 |
| F58A | > 2000 |
| R59E | 9.3 ± 3.3 |
| ΔKIR | > 2000 |
| D105A | 30 ± 20 |
| F112A | 1950 ± 170 |
| JAK1 GQM-DVP | > 2,000 |

**e** Kinase inhibitory region

|  | 54 |  |  |  |  |  |  |  | 63 |
|---|---|---|---|---|---|---|---|---|---|
| Human | H F R T F R S H A D |
| *M. musculus* | H F R T F R S H S D |
| *G. gallus* | H F R T F R S Q A D |
| *X. laevis* | H F R T F R S H S D |
| *D. rerio* | H F K P F N Y A R D |

**Fig. 3** The kinase inhibitory region acts as a pseudosubstrate. **a** Molecular details of the interaction between the SOCS1 kinase inhibitory region (red) and JAK1 (beige). The interaction is driven by six continuous residues in the SOCS1 KIR (His54 to Arg59), these primarily interact with the activation loop, GQM motif and glycine-rich loop of JAK1. Black dashed lines indicate hydrogen-bonds whilst grey dotted lines are van der Waals contacts. **b** A model of a peptide substrate (green) bound to the JAK1 kinase domain (based on the insulin receptor kinase/substrate structure (PDB: 1IRK)) indicates that the SOCS1 KIR acts as a pseudosubstrate with His54 mimicking the substrate tyrosine. **c** The GQM motif (Gly1097, Gln1098, Met1099) in JAK1 is a primary interaction site with SOCS1 and contacts the KIR, extended SH2 subdomain and SH2 domain (particularly the BC loop). **d** IC$_{50}$ values of SOCS1 KIR mutants indicate that each residue of the KIR contributes to the affinity of the interaction. Errors are standard error of the mean from two independent experiments. **e** Sequence alignment of the KIR of SOCS1 from various species shows a high level of sequence conservation

and Phe113. These interactions are facilitated by the lack of a sidechain at Gly1097, which allows close approach of the BC loop. In order to determine the importance of the GQM/BC loop interaction, several mutations were made and the effects of these examined via kinase inhibition assays. As shown in Fig. 3d, mutation of Asp105 and Phe112 increased the IC$_{50}$ against JAK1 30 and 2000-fold, respectively, and mutation of the GQM motif in JAK1 abolished the interaction entirely.

**SOCS1 binds to unphosphorylated JAK1.** When bound to SOCS1, the JAK1 kinase domain adopts an active conformation, characterised by the activation loop displaced from the ATP-binding site with a DFG-in position, an intact substrate-binding groove and both the C-spine and R-spine[26] aligned. This suggests that, if the KIR of SOCS1 was not blocking the substrate-binding

groove, the enzyme would be catalytically active. ADP is bound in the active site of the kinase, representing the hydrolysed form of the ATP present in the crystallisation solution. The activation loop is in its active conformation and this is notable because Tyr1034 is unphosphorylated. Autophosphorylation of Tyr1034 (in *trans*) is the key event required for the kinase domains of JAKs to become activated[27]. We observed partial (estimated at ~20%) phosphorylation of the adjacent tyrosine (Tyr1035), however, this tyrosine is not required or sufficient for activation. Interestingly, while SOCS1 is bound to JAK1, not only is the bound JAK1 prevented from *trans*-phosphorylating a second JAK1 molecule (Supplementary Figure 2A) but also appears blocked from being trans-phosphorylated itself. This is because the key tyrosine cannot access the active site of a second kinase molecule while locked in its open (active) conformation[28,29], and also because steric hindrance from SOCS1 itself would prevent

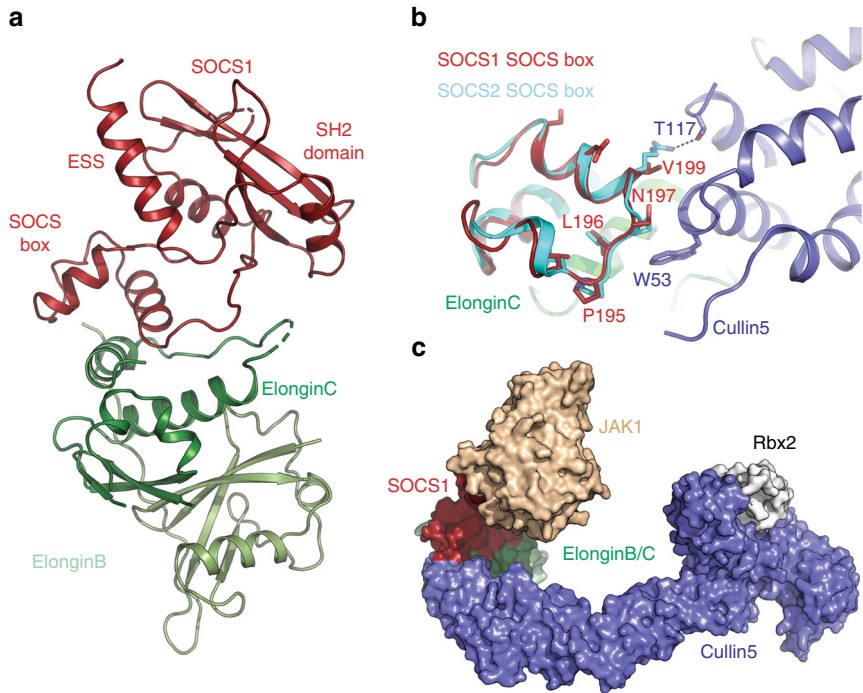

**Fig. 4** Structure of the SOCS1–Elongin B/C complex. **a** Structure of *Xenopus laevis* SOCS1 (red) in complex with Elongin B (pale green) and Elongin C (green). **b** The SOCS1 SOCS Box is shown overlaid with the structure of SOCS2 (cyan) bound to Cullin5 (blue). SOCS1 Asn197 takes the place of terminal proline typically seen in the canonical Cullin5 binding motif (LPφP). This asparagine clashes with Trp53 of Cullin5. Likewise, the Arg186 (SOCS2)-Thr117 (Cul5) hydrogen bond is lost in SOCS1 which has a valine (Val200) in place of the arginine. **c** Model of the JAK1-SOCS1-Elongin B/C-Cullin5-Rbx2 (white) complex based upon the SOCS1/JAK1 (this manuscript), SOCS2/ElonginB/ElonginC/Cullin5 NTD (PDB: 4JGH) and Cul5 CTD/Rbx2 structures (PDB: 3DPL). The SOCS1/JAK1 interaction orients JAK1 toward rbx2, which is the site of activated-ubiquitin (ubiquitin-E2) binding

the close approach of the second kinase molecule (Supplementary Figure 2B). Therefore, SOCS1 binding blocks the activation of JAK1 as well as blocking the catalytic activity of its activated form.

**The SOCS1 SOCS box prevents high affinity recruitment of Cullin5.** In addition to kinase inhibition, SOCS1 can recruit an E3 ubiquitin ligase to regulate signalling. However, SOCS1 binds only weakly to the E3 ligase scaffold protein Cullin5, with 100 times lower affinity than the majority of the SOCS family. This lower affinity is due to an atypical Cul5 binding sequence (IPLN) within the SOCS1 SOCS box domain[30], which lacks the strictly conserved final proline of the canonical LPφP motif. We solved the structure of SOCS1 from *Xenopus laevis* in complex with Elongins B and C (Table 1). Although the human form of the complex did not crystallise, the SOCS box from *xl*SOCS1 is 73% identical and 85% similar to the human domain and contains identical Elongin BC and Cullin5 binding motifs.

SOCS1 forms a complex with Elongin BC that is structurally similar to other SOCS/Elongin BC complexes (Fig. 4a). Comparison of the structure of the SOCS box with the same domain from SOCS2 (which targets Cullin5 with 100-fold higher affinity) shows no large-scale structural differences between the two. Overlay of the Cullin5 binding region from SOCS1 and SOCS2 indicates that the IPLN sequence in SOCS1 occupies roughly the same spatial position as the canonical LPφP motif in SOCS2 (Fig. 4b). This asparagine in SOCS1 overlays the orthologous proline in the SOCS2 motif, however, cannot form the same ring-stacking interaction with Trp53 of Cul5[31]. The only other residue in SOCS2 that makes contact with Cullin5 is Arg186, two-residues downstream of the LPφP motif. This residue is Val199 in SOCS1 and has too short a sidechain to make contact with Cul5. Thus, there are no residues in SOCS1 that make favourable contacts with Cullin5, and the interaction between SOCS1/BC

and Cullin5 is likely mediated purely by the ElonginC–Cullin5 interaction alone.

The structure of SOCS1/Elongin BC can be combined with the SOCS1/JAK1 and SOCS2/Elongin BC/Cullin5 structures to generate a model of the full E3 ligase–Substrate complex (Fig. 4c). When the KIR of SOCS1 is occupying the substrate-binding groove of JAK1, it orients the JAK1 protein correctly towards the E2-Ubiquitin docking site on Rbx2. The distance between the JAK1 kinase domain and Rbx2 is 60 Å, a distance that would be covered in the context of full-length JAK1 especially in the presence of Cullin5 neddylation as shown previously[32].

**SOCS1 does not bind to the interferon gamma (IFNγ) receptor.** SOCS proteins are thought to gain specificity for particular cytokines by using their SH2 domains to bind to phosphotyrosine (pTyr) motifs within the cytoplasmic domain of certain cytokine receptors[22,24]. Therefore, we chose the three most well-characterised targets of SOCS1 activity in vivo[33] (IFN-γ, IFN-α and IL-2 family cytokines) and investigated their receptors for potential SOCS1 binding sites in vitro. An analysis such as this does not prove that such an interaction occurs in vivo, however, it is a powerful method for showing which interactions do not occur.

We obtained synthetic phosphopeptides representing every potential pTyr site within the cytoplasmic domains of these receptors (30 in total) (Fig. 5a, Table S1) and tested whether SOCS1 could bind these peptides using isothermal titration calorimetry (ITC). Surprisingly, human SOCS1 could not bind to any pTyrs from the IFN-γ receptor even though IFN-γ is the primary target of SOCS1 action (Fig. 5b, c). Four sites on the IL-2 receptor interacted with biologically relevant affinity (sub-micromolar) however these were located within the beta chain. As only IL-2/15, and not other IL-2 family members signal via the beta chain, the relevance of this interaction is unclear.

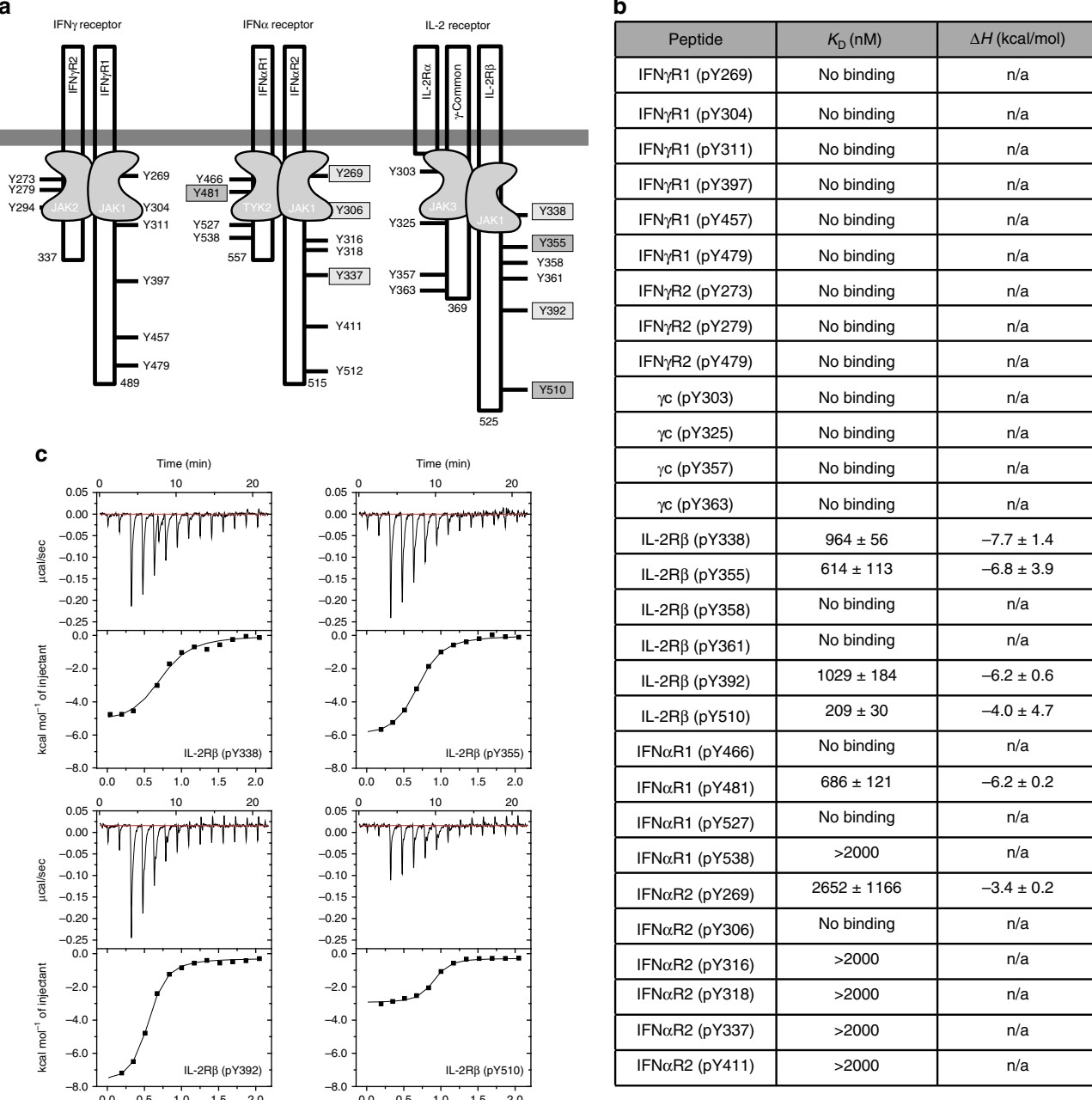

**Fig. 5** SOCS1 does not bind to any pTyr sites on the interferon gamma receptor. **a** Schematic diagrams of the IFN-α, ɣ and IL-2 receptors. All intracellular tyrosines are shown. **b** Table showing the binding affinities of human SOCS1 for each IFN-ɣ and IL-2 receptor phosphopeptide. Errors represent standard deviation from three independent experiments. **c** Representative ITC curves are shown for the four IL-2R phosphopeptides that bind to SOCS1 with sub-micromolar affinity

Finally, although SOCS1 was able to bind to one peptide from the IFN-α receptor with sub-micromolar affinity, this site is within the Box1/2 motif and is occupied by TYK2 in vivo (Fig. 5b, c)[34]. These data show that, for interferon-ɣ, and potentially other cytokines, SOCS1 does not directly target the receptor via its SH2 domain.

**The SOCS1 SH2 domain binds JAK activation loop peptides.** In their initial characterisation of SOCS1, Yoshimura and colleagues postulated that the SOCS1 SH2 domain binds the activation loop of JAKs[4]. As the activation loop of JAK1 was not phosphorylated in our SOCS1/JAK1 structure, such an interaction could not occur. Therefore, we obtained synthetic peptides representing the activation loop of all four JAKs to investigate whether such an interaction is possible. As shown in Fig. 6, SOCS1 bound tightly (<1 µM affinity) to activation loops phosphopeptides of all four JAK family members. As the interaction with the JAK1 activation loop was higher than any receptor peptides, we generated a series of alanine point mutants of this peptide to attempt to derive a consensus sequence (Supplementary Table 2). Although no definitive consensus motif could be identified, analysis of these data in addition to the binding data from the receptor peptides suggest that hydrophobic residues at the +1 and +3 positions are preferred (pY–ϕ–X–ϕ).

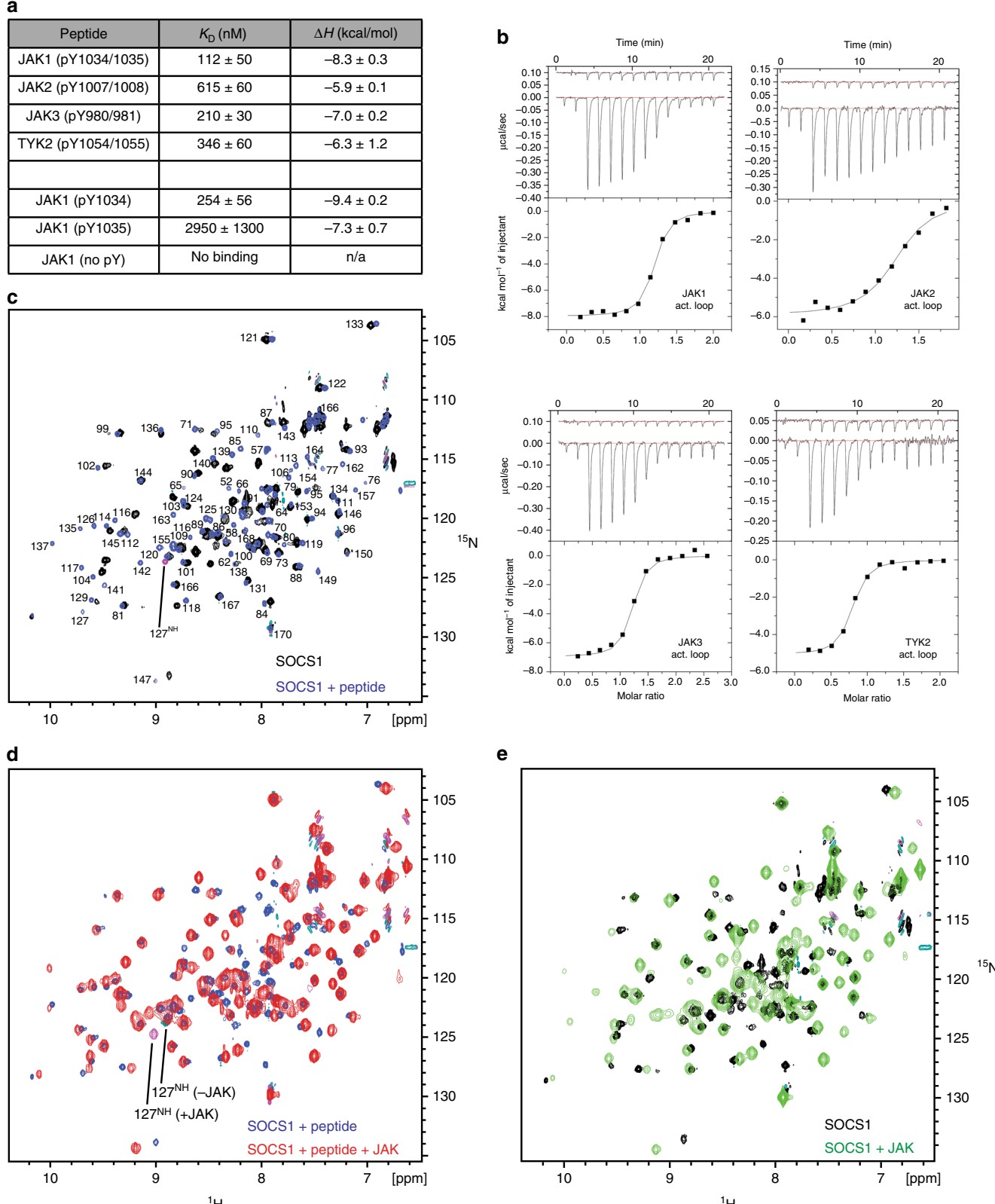

**a**

| Peptide | $K_D$ (nM) | $\Delta H$ (kcal/mol) |
|---|---|---|
| JAK1 (pY1034/1035) | 112 ± 50 | −8.3 ± 0.3 |
| JAK2 (pY1007/1008) | 615 ± 60 | −5.9 ± 0.1 |
| JAK3 (pY980/981) | 210 ± 30 | −7.0 ± 0.2 |
| TYK2 (pY1054/1055) | 346 ± 60 | −6.3 ± 1.2 |
| | | |
| JAK1 (pY1034) | 254 ± 56 | −9.4 ± 0.2 |
| JAK1 (pY1035) | 2950 ± 1300 | −7.3 ± 0.7 |
| JAK1 (no pY) | No binding | n/a |

**SOCS1 can bind JAK and a pTyr peptide simultaneously.**
Although SOCS1 could bind activation-loop-derived phospho-peptides with high affinity, our attempts to crystallise a SOCS1/JAK1 complex bound to a pTyr peptide were unsuccessful (possibly due to crystal packing preventing a phosphopeptide from binding the SOCS1 SH2 domain, Supplementary Figure 3).

To determine whether SOCS1 can bind JAK (via its KIR) and simultaneously bind a phosphopeptide via its SH2 domain, NMR studies were undertaken. $^{13}C/^{15}N$-labelled $gg$SOCS1$^{\Delta SOCSbox}$ was expressed and purified and the resonances assigned. As shown in Fig. 6c, there was a large-scale perturbation of the $^{15}N$-HMQC spectra upon addition of an unlabelled JAK1 activation loop

**Fig. 6** SOCS1 binds to the activation loop peptides from all four JAKs. **a** Table showing the affinity and enthalpy of each SOCS1/JAK activation loop peptide interaction as determined by isothermal titration calorimetry. Human SOCS1 binds to phosphorylated peptides representing all four JAK activation loops. The first phosphotyrosine of the JAK1 activation loop (pTyr1034) is the key residue for SOCS1 binding. Errors represent standard error of the mean from three independent experiments. **b** Representative ITC binding data for the four JAK activation loop peptides (doubly-phosphorylated) binding to SOCS1 used to generate **a**. **c** NMR analysis of phosphopeptide binding. $^1$H–$^{15}$N SOFAST HMQC spectra of ggSOCS1 in the presence (blue) and absence (black) of the JAK1 activation loop are shown overlaid. Both spectra were assigned, the labels indicate the positions of amide resonances from the bound form of the protein. **d** NMR analysis of JAK binding in the presence of exogenous phosphopeptide. $^1$H–$^{15}$N SOFAST HMQC spectra of ggSOCS1/activation loop complex in the presence (red) and absence (blue) of the JAK1 kinase domain are shown overlaid. The sidechain epsilon NH resonance of Arg127 from both JAK-bound and unbound forms are indicated. These are of opposite phase to backbone amide resonances. The presence of the Arg127 resonance in that section of spectra indicates that the phosphopeptide-binding groove is occupied in both complexes. **e** NMR analysis of JAK binding in the absence of exogenous phosphopeptide. $^1$H–$^{15}$N SOFAST HMQC spectra of ggSOCS1 in the presence (green) and absence (black) of the JAK1 kinase domain are shown overlaid. The line-broadening in the green spectra indicated the formation of a SOCS1/JAK complex and the lack of the Arg127 sidechain resonance in either spectra indicates the phosphotyrosine-binding groove is unoccupied

phosphopeptide. The resonances were in slow exchange upon phosphopeptide binding, and therefore separate assignments were made for the bound form. An analysis of the magnitude of the chemical shift perturbation for each residue suggests that the peptide occupied the canonical pTyr groove of the SOCS1 SH2 domain. Of special interest was the resonance position of side-chain epsilon NH moiety from Arg127, which has an unusual chemical shift when bound to the phosphopeptide but is unresolved in the apo-form. This residue assists in co-ordinating the phosphate moiety of the pTyr residue in many structures of SH2 domains bound to their targets and its appearance in the SOCS1/phosphopeptide spectra could be used as a signature for SH2 domain occupation by pTyr. As shown in Fig. 6d, upon titration of the SOCS1/activation loop peptide complex with fully activated JAK2 kinase domain, there were numerous further shifts and significant line broadening, indicating the formation of a SOCS1/JAK2 complex. Importantly, the resonance from the Arg127 sidechain was clearly visible. This indicates that the peptide remained bound and that a ternary JAK2/SOCS1/peptide complex was formed. There were other characteristic resonances that also supported this hypothesis. Taken together, NMR analysis showed that SOCS1 can bind to JAK in the presence and absence of phosphopeptide and that the phosphopeptide is not displaced upon JAK binding and vice-versa. Interestingly, the Arg127 sidechain resonance was not visible when a JAK2/SOCS1 complex was formed in the absence of phosphopeptide (Fig. 6e), indicating that the SH2 domain was not occupied by phosphotyrosine in this complex, even though the JAK2 kinase domain was fully phosphorylated.

**Activation loop binding is sterically hindered by the kinase domain**. Although SOCS1 bound the activation loop (as a peptide) with high affinity, the NMR analysis described above did not support a model in which a phosphotyrosine from the activation loop occupied the SOCS1 SH2 domain. Additionally, all JAK structures to date suggest that the two pTyrs within this loop would be shielded from binding an SH2 domain in the canonical manner by the rest of the kinase domain. Therefore, we undertook a series of experiments to determine whether SOCS1 could bind the activation loop when it was part of an intact kinase domain. JAK2 was chosen as the basis for most of these experiments as we found it could be quantitatively autophosphorylated by incubation with ATP, unlike JAK1. Recombinant human and ggSOCS1$^{\Delta KIR}$ (in order to abrogate the KIR-based interaction) were then mixed with phosphorylated JAK2 kinase domain (JAK2$^{KD}$) and complex formation was analysed.

Surprisingly, no binding was detected between SOCS1$^{\Delta KIR}$ and JAK2$^{KD}$ by ITC, gel filtration or NMR (Fig. 7a, b, e). As a control, both species of SOCS1$^{\Delta KIR}$ retained the ability to bind the phosphorylated JAK2 activation loop as a peptide. To further

confirm a lack of interaction, kinase inhibition assays, using wild-type SOCS1, were performed in the presence of competing phosphopeptide. As shown in Fig. 7c, the presence of 180 μM phosphorylated activation loop peptide (300-fold above $K_d$) did not alter the IC$_{50}$ of SOCS1. If the activation loop was bound by SOCS1, competing this interaction with activation loop peptide would be expected to lead to decreased affinity and therefore an increased IC$_{50}$. As a positive control for this experiment, we created a JAK construct in which the activation loop would be accessible for binding by fusing the activation loop sequence to the unstructured N-terminus of the JAK kinase domain (ALP-JAK1). ALP-JAK1 was inhibited with extremely high (sub-nanomolar) affinity by SOCS1, presumably because SOCS1 could bind this artificial construct using both its KIR and SH2 domain. This artificial JAK construct also formed a stable complex with SOCS1$^{\Delta KIR}$ as expected (Fig. 7b).

Taken together, our data indicate that the SH2 domain of SOCS1 binds the phosphorylated activation loop sequence with high affinity but cannot do so in the context of an intact JAK kinase domain because it is sterically hindered. Interestingly, we did observe evidence of a weak SOCS1$^{\Delta KIR}$/JAK complex by gel filtration when JAK was bound by a type II inhibitor (Fig. 7d). Type II inhibitors are known to induce a disordered activation loop[35] which we hypothesized may allow it to be targeted by the SOCS1 SH2 domain with less interference from the kinase domain proper. Although the affinity of this complex was too low to quantify, it may suggest that, in vivo, any events that release the activation loop could remove the steric hindrance that otherwise blocks association with the SOCS1 SH2 domain.

## Discussion

SOCS1 was first identified as a JAK-binding inhibitor of cytokine signalling[5–7]. In 1999, a landmark paper by Yoshimura and colleagues identified sub-domains within the protein that are crucial for function and proposed a model for its mode-of-action[4]. Their key hypotheses were that the SH2 domain of SOCS1 (which includes an unusual N-terminal extension) can bind to a specific phosphotyrosine in the activation loop of JAK1 and JAK2, and that a kinase inhibitory region near the N-terminus of the protein then allowed it to directly inhibit the catalytic activity of the kinase.

Here we have shown that SOCS1 inhibits JAK by using its KIR to block the substrate-binding groove of the kinase. The N-terminal residue of the KIR is a histidine, which occupies the substrate tyrosine-binding site on JAK1. Another kinase-inhibitory protein, Mig6 (an EGFR inhibitor) acts as a substrate competitor and also places a histidine sidechain in the substrate-biding site[36]. The kinase inhibitory segment of Mig6 adopts a beta-hairpin configuration and is anchored to its target kinase by a distal element upstream of it. The KIR of SOCS1 is also

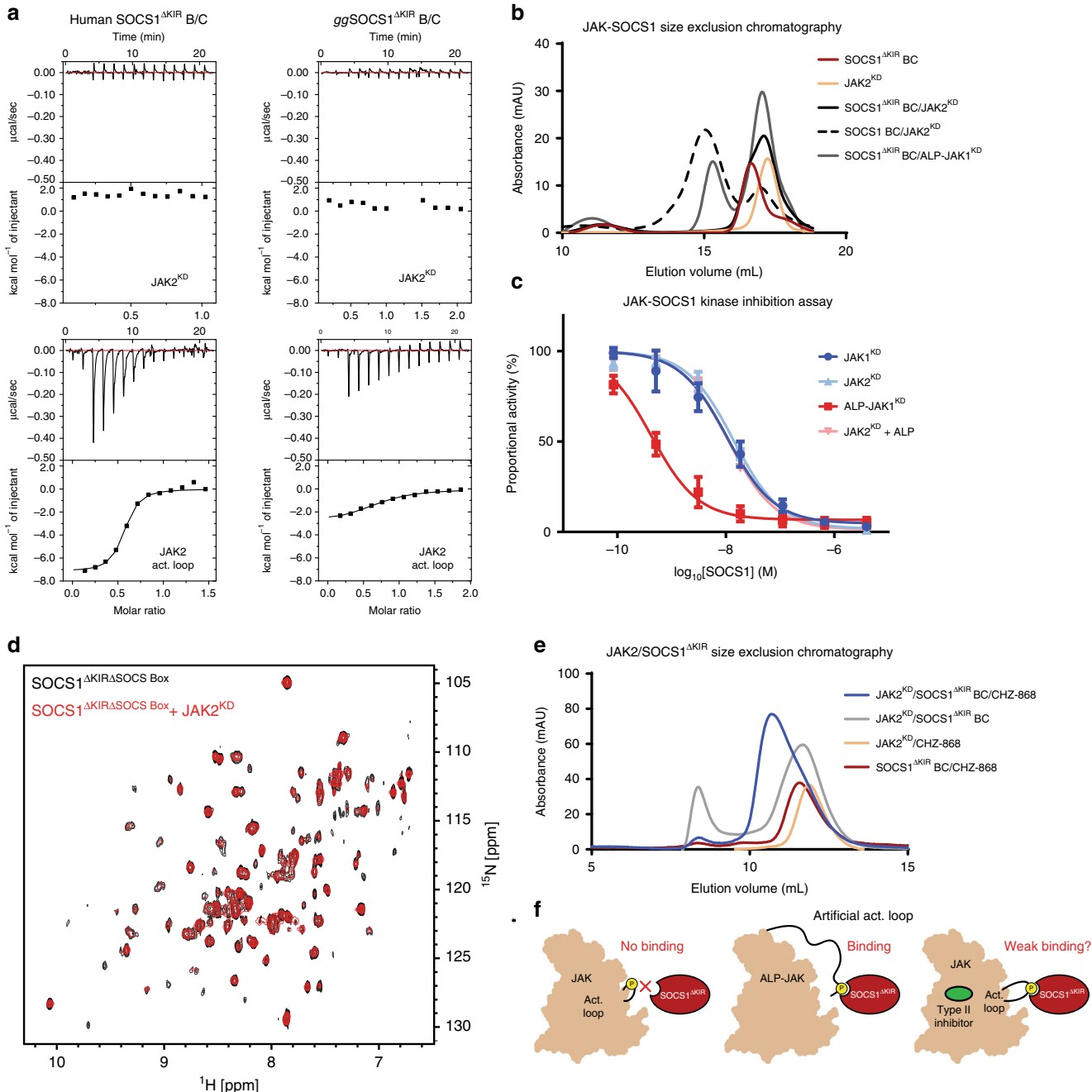

**Fig. 7** Steric hindrance prevents SOCS1 from binding the JAK activation loop in the intact kinase domain. **a** ITC binding data indicates both human SOCS1$^{\Delta KIR}$ B/C (left) and *gg*SOCS1$^{\Delta KIR}$ B/C (right) do not bind to the JAK2 kinase domain but do bind to JAK2 activation loop peptides. **b** Size exclusion chromatography (using a Superdex 200 10/300 column) data indicate that SOCS1$^{\Delta KIR}$ B/C does not form a complex with JAK2. As positive controls, wild-type SOCS1 B/C does form a complex with JAK2, while SOCS1$^{\Delta KIR}$ B/C does form a complex with JAK1 when the activation loop peptide (ALP) motif is artificially fused to the N-terminus of JAK. (ALP-JAK1$^{KD}$). **c** Kinase inhibition assays show that SOCS1 inhibits JAK2 with similar IC$_{50}$ values in the absence and presence of activation loop peptide. Conversely, SOCS1 displays a lower IC$_{50}$ against ALP-JAK1$^{KD}$. Error bars represent the range of the data from two technical replicates. **d** NMR data indicate that *gg*SOCS1$^{\Delta KIR \Delta SOCSbox}$ does not bind the phosphorylated JAK2 kinase domain. $^{1}$H–$^{15}$N SOFAST HMQC spectra of 15N-labelled *gg*SOCS1$^{\Delta KIR \Delta SOCSbox}$ in the presence (red) and absence (black) of the phosphorylated JAK2 kinase domain (unlabelled) are shown overlaid. **e** Size exclusion chromatography data (using a Superdex 75 10/30 column) indicate that SOCS1$^{\Delta KIR}$ B/C forms a complex with JAK2 in the presence of 5 µM CHZ-868 Type II inhibitor, but not in its absence. **f** Schematic summary of SOCS1 SH2 domain interactions. Under normal conditions, SOCS1$^{\Delta KIR}$ does not bind to the JAK activation loop due to steric hinderance. In the presence of a Type II inhibitor, the SOCS1 SH2 domain weakly associates with the JAK activation loop

anchored to its target by regions of the protein distal to it in the primary sequence and it seems likely that both proteins require a higher-affinity interaction with their targets than can be provided by the kinase inhibitory segments alone. Both SOCS3 (a closely related inhibitor of JAK)[22] and Grb14[37] (an inhibitor of the insulin receptor kinase) obey similar rules, however, these proteins use an arginine and a leucine, respectively, to mimic the substrate tyrosine, indicating the structural plasticity available to this class of kinase inhibitor. Nevertheless, it is clear that the KIR of SOCS1 is highly evolved to target the substrate-binding groove of JAK, as mutating any of the six continuous residues that contact this groove, which are highly conserved in all orthologues, led to a significant decrease in affinity.

The presence of two binding surfaces on SOCS1 (KIR and BC loop) acting simultaneously, leads to a high-affinity interaction between the protein and its target. SOCS1 is a particularly potent inhibitor of JAK1 and JAK2, inhibiting with low nanomolar IC$_{50}$. Notably, SOCS1 is an order of magnitude more potent at inhibiting JAK than is SOCS3. Indeed, by virtue of inhibiting JAK via a mechanism that is not ATP-competitive, SOCS1 approaches the same potency as Ruxolitinib[38] in the presence of intracellular levels of ATP (ca. 5–10 mM). A second, and important, aspect to the potency of SOCS1 is that, as shown here structurally, it can bind to dephosphorylated JAK and does so in such a way that it locks the activation loop. As such, the activation loop can no longer act as a substrate in trans and bind to the active site of a second kinase molecule[29]. The ability of SOCS1 to bind with such high affinity to dephosphorylated JAK, without the aid of a phosphorylated cytokine receptor may explain why, in vivo SOCS1 can inhibit JAK autophosphorylation (as well as downstream phosphorylation) in contrast to SOCS3 which cannot[4].

While SOCS1 has potent direct inhibitory activity against its JAK targets it has reduced ability to target Cullin5 (and thereby induce ubiquitination). The affinity of SOCS1 for Cullin5, the E3 ligase scaffold protein recruited by all SOCS proteins, is 100-fold lower than the other members of the SOCS family[30]. The structure of SOCS1 in complex with Elongin BC explains this decrease in affinity. Consistent with these findings, genetic deletion of the SOCS1 SOCS box domain[39] did not recapitulate the neonatally lethal phenotype of the full SOCS1 knockout[8]. Nevertheless, these mice have increased responsiveness toward IFN-γ and slowly develop a lethal inflammatory disease[39], indicating that while direct kinase inhibition is its major mode-of-action in vivo, ubiquitination still plays a role.

Although SOCS proteins can inhibit signalling by numerous cytokines when overexpressed, knockout studies have shown that they are highly specific under physiological conditions. In the case of SOCS3, this specificity is achieved by using its SH2 domain to target particular phosphotyrosine motifs on certain receptors, while simultaneously inhibiting the associated JAK molecule with its KIR. As expected, these motifs are highly evolutionarily conserved. Here we showed that SOCS1 could not target any phosphotyrosine-based motifs on the IFN-γ receptor despite being a potent inhibitor of the associated cytokine[9]. Although there was one high-affinity site on the human IFN-α receptor that SOCS1 could bind to it is not well conserved. Indeed, it has previously been shown that no tyrosines within the interferon alpha receptor 1 chain are absolutely required for inhibition by SOCS1[40]. Likewise, receptor binding is not required for SOCS1 inhibition of IL-2[41]. Therefore, although SOCS1 can bind to certain receptor motifs in vitro, data shown here and by others suggest SOCS1 does not require receptor-binding in vivo. This makes its mode-of-action distinct from SOCS3. It may be that, by bypassing receptor involvement, SOCS1 is better able to inhibit a wide range of cytokines and indeed SOCS1 targets more cytokines than any member of the SOCS family.

If SOCS1 does not utilise receptor recruitment as part of its mechanism-of-action then how can it achieve specificity for some cytokines but not others? It is possible that SOCS1 specificity is purely driven by the relative strength of its induction by each cytokine as suggested previously[42]. However, there are some cytokines, such as IL-6, whose regulation is not affected by SOCS1 knockout[43,44] even though SOCS1 is induced by those cytokines and is capable of inhibiting them when artificially overexpressed. It seems likely that SOCS1 specificity must arise, mechanistically, from its SH2 domain. Here we showed that SOCS1 could bind the activation loop (as an isolated peptide) of all four JAK family members with high affinity. However, it was sterically hindered from doing so when these activation loops were part of an intact kinase domain. Nevertheless, it is clear that the activation loop is a mobile element[25,29,45] (Supplementary Figure 4) and it is conceivable that, in vivo, interactions with other proteins may induce a conformational change in the activation loop such that it becomes accessible for SOCS1 binding (Supplementary Figure 5). It is notable that the two loops within the SOCS1 SH2 domain that form the pY + 3 pocket (the specificity-determining pocket within most SH2 domains) are the shortest of any SH2 domain in the human proteome, to our knowledge. Such a feature would be required if SOCS1 was to bind the activation loop and avoid those loops clashing with the kinase domain proper, as opposed to binding an unstructured linear receptor motif.

The vast majority of kinase inhibitors are ATP-competitive, targeting the ATP-binding site of their kinase. Inhibitors that act via a different mechanism, targeting a site on the kinase that is distal to ATP can be beneficial for a number of reasons. For example, such a molecule would not be required to compete with high concentrations of ATP found inside the cell, improving efficacy. More importantly, such an inhibitor may be more specific for a single kinase because the ATP-binding pocket is the most structurally conserved region throughout the kinome. It is becoming clear that Type I JAK inhibitors, while being effective in treating the symptoms of JAK2$^{V617F}$-driven myeloproliferative disease, do not cure the disease. However, there has been recent success using non-type I JAK inhibitors in preclinical trials[46]. The kinase inhibitory region that is shared by SOCS1 and SOCS3 does not act as a type I inhibitor. The atomic-resolution details of the kinase inhibitory region of SOCS1 provides the structural basis for designing a small-molecule mimetic of the KIR in order to develop a new class of JAK inhibitor. Conversely, recent studies have shown that active interferon-γ signalling is required for cancer immunotherapy approaches, including PD-1 blockade, to be effective[47–49]. Therefore, pharmacological inhibition of SOCS1 should increase the effectiveness of these types of cancer immunotherapies and the structures of SOCS1 presented here will enable structure-guided drug design approaches towards this aim.

## Methods

**Cloning expression and purification of SOCS1 and JAKs.** Human SOCS1 (residues 53–211), *Gallus gallus* SOCS1 (residues 48–207) and *Xenopus laevis* SOCS1 (residues 53–211) were cloned into pGEX4T (eloB)[50] using AscI and EcoR1 cloning sites for use in inhibition assays. A TEV site was incorporated at the N-terminus of SOCS1 to allow the removal of the GST tag. This vector also encoded untagged ElonginB. This vector was co-transformed with pBB75 (human ElonginC 17–112)[50] into BL21(DE3) cells and expressed by 1 mM IPTG induction at 18 °C overnight to produce ternary SOCS1/ElonginB/ElonginC complexes. Cells were colleccted by centrifugation and frozen at −30 °C. Cells from 1 L culture were resuspended in 50 mL SOCS1 Purification Buffer [10 mM Tris-HCl pH 7.5, 150 mM NaCl, 5 mM β-mercaptoethanol, 5 mM phenylphosphate] plus 1 mM phenyl-

methylsulfonyl fluoride (PMSF), protease inhibitor cocktail (Roche), 250 U DNASe, and lysed by sonication. The lysate was cleared by centrifugation at 50,000 × g for 30 min and bound to 1 mL GSH-agarose resin (GE lifesciences) by gravity flow. The resin was washed with 50 mL SOCS1 Purification Buffer. TEV cleavage was performed to release SOCS1 from bound GST, followed by size exclusion chromatography on a Superdex 200 26/600 column (GE Healthcare) in SOCS1 Purification Buffer. In order to remove the phenylphosphate SOCS1 was buffer exchanged into 10 mM Tris pH 7.5, 150 mM NaCl, 2 mM β-mercaptoethanol using a PD-10 desalting column (GE Healthcare) immediately prior to assays. Human JAK1, JAK2, JAK3 and TYK2 kinase domains (KD) were cloned into pFASTBAC (Invitrogen), expressed as 6xHis-tagged proteins and purified as previously described[24].

**JAK kinase inhibition assays using peptide substrates**. A concentration of 0–2 mM substrate peptide (derived from STAT5b) was incubated with 5–10 nM JAK1$^{KD}$, JAK2$^{KD}$, JAK3$^{KD}$ or TYK2$^{KD}$ with varying concentrations of SOCS proteins at 25 °C for 10–30 min in 20 mM Tris pH 7.5, 150 mM NaCl, 2 mM β-mercaptoethanol, 0.1 mg/mL BSA, 2 mM MgCl$_2$ 1 mM ATP supplemented with 1 μCi $^{32}$P-γ-ATP. Following this, the reactions were spotted onto P81 phosphocellulose ion-exchange paper and quenched in 5% H$_3$PO$_4$. The paper was washed (4 × 200 ml, 15 mins) with 5% H$_3$PO$_4$, dried and then and exposed to a phosphorimager plate (Fuji).

**Cloning expression and purification of JAK1-SOCS1 complex**. *Gallus gallus* SOCS1 (residues 48–164) was cloned into pACEBac1 (ATG:biosynthetics GmbH) using HindIII and StuI cloning sites. A C-terminal precision protease site was introduced followed by streptavidin and S-protein tags. Purified plasmids were used to make bacmid and transformed into baculovirus using standard techniques. *Spodoptera frugiperda* 21 (Sf21) cells were infected with baculoviruses encoding SOCS1 and JAK1 (JAK1 constructs as described above) at multiplicities of infection of 2 and 3, respectively. The cells were collected 48 h post infection by centrifugation and frozen at −30 °C. Frozen cells from 1 L of cell culture were thawed and resuspended in 100 mL JAK-SOCS Lysis Buffer [10 mM Tris-HCl pH 7.5, 150 mM NaCl, 2 mM TCEP, 1 mM phenyl- methylsulfonyl fluoride (PMSF), protease inhibitor cocktail (Roche), 250 U DNASe]. Cells were lysed by sonication and the lysate was cleared by centrifugation for 60 min at 50,000 × g. Cleared lysate was filtered through a 0.8 μm filter and loaded onto a 1 mL HisTrap HP affinity column. The column was washed with Nickel Buffer A [20% glycerol, 20 mM Tris-HCl pH 8.0, 500 mM NaCl, 5 mM imidazole, 2 mM TCEP] followed by 98% (v/v) Nickel Buffer A with 2% (v/v) Nickel Buffer B [20% glycerol, 20 mM Tris-HCl pH 8.0, 500 mM NaCl, 500 mM imidazole, 2 mM TCEP] and then 93% (v/v) Nickel Buffer A with 7% (v/v) Nickel Buffer B. The protein was eluted from the column with a 7–100% gradient of Nickel Buffer A – Nickel Buffer B. Tags were cleaved overnight at 4 °C with TEV protease and precision protease. JAK1–SOCS1 complex was concentrated to <4 mL and loaded onto a Superdex 200 16/600 size exclusion column in JAK Gel Filtration Buffer [10% glycerol, 20 mM Tris-HCl pH 8.0, 500 mM NaCl, 2 mM TCEP]. Fractions were assayed by reducing SDS-PAGE gel and those containing pure complex (>95%) were pooled, concentrated, and buffer components were adjusted to 10 mM Tris-HCl pH 8.0, 100 mM NaCl, 3% glycerol, 2 mM TCEP, 1 mM ATP, 2 mM MgCl$_2$.

**Crystallisation and structure determination of JAK1/SOCS1**. Crystallization was accomplished by hanging-drop vapour diffusion at 8 °C, using protein at 7 mg/ml, and a drop ratio of 1:1 protein: precipitant. Precipitant conditions were optimised to 100 mM HEPES pH 7.0, 14% (w/v) PEG 8000, 100 mM MgAc, 2 mM TCEP. Crystals were flash-frozen in liquid nitrogen, using mother liquor with an additional 18% (v/v) ethylene glycol as cryoprotectant. Diffraction data were collected on beamline MX2 at the Australian Synchrotron using a wavelength of 0.9537 Å. Data were integrated using XDS[51] and scaled using XSCALE[51]. The data used for refinement was cut at 2.5 Å. The data in the highest resolution shell had an I/σ of 1.90 and a CC½ of 69.1% (Table 1). A molecular replacement solution was obtained with PHASER[52], using *Xenopus laevis* SOCS1 (PDB ID: 6C5X) and JAK1 (PDB ID: 3EYG) as search models with one copy of each molecule in the asymmetric unit. Refinement was performed using PHENIX[53] and model building was performed in COOT[54]. Refinement converged to $R_{work}$ and $R_{free}$ values of 0.2055 and 0.2414 respectively. The final refined model had 96.6% residues in the favoured region and 3.4% residues in the allowed region of the Ramachandran plot.

**Crystallisation and structure determination of SOCS1/ElonginBC**. In order to determine the structure of a SOCS1/ElonginB/ElonginC ternary complex, SOCS1 from all three species that could be successfully expressed and purified were tested in high-throughput crystallization trials. A variety of different phosphopeptides were added to aid stabilization of the SH2 domain. Crystals were successfully obtained only of a complex between SOCS1 from *Xenopus laevis* with human elonginB and elonginC in the presence of a low-affinity phosphopeptide from the murine gp130 receptor. This phosphopeptide is unlikely to represent a

physiological target of the *Xenopus* SOCS1 SH2 domain and was included solely to aid stabilization and crystallisation. The complex did not crystallise in the absence of this peptide, nor in the presence of higher-affinity phosphopeptides. Crystallization was accomplished by hanging-drop vapour diffusion at 20 °C, using protein at 10 mg/mL and a drop ratio of 1:1 protein: precipitant. Crystals of various sizes were obtained from 18% PEG-3350, 200 mM NaF, 100 mM Tris 7.5. Crystals were flash-frozen in liquid nitrogen, using paratone as a cryoprotectant. Diffraction data were collected on beamline MX2 at the Australian Synchrotron using a wavelength of 0.9537 Å. Data were integrated using XDS[51] and scaled using XSCALE. The data for refinement was cut at 3.1 Å. The data in the highest resolution shell had an I/σ of 1.54 and a CC½ of 0.577 (Table 1). The cutoff for the data used in refinement was determined using the Pearson correlation coefficient, as represented in the XSCALE output[55]. The structure was solved by molecular replacement using the three chains of PDB ID 2C9W (SOCS2/ElonginB/ElonginC) as individual search models. Refinement converged to $R_{work}$ and $R_{free}$ values of 0.2353 and 0.2714, respectively. The final refined model had 96% residues in the favoured region and 3.5% residues in the allowed region of the Ramachandran plot.

**NMR analysis**. ggSOCS1$^{ΔSOCSbox}$ (*Gallus gallus* SOCS1 residues 48–166) was cloned into pET15b as a 6xHIS fusion protein. The protein was then expressed overnight at 25 °C in BL21(DE3) cells in Neidhardts minimal media[56] with 1 g/L 15 N NH$_4$Cl and 4 g/L 13C-glucose as the nitrogen and carbon sources, respectively. The 6xHIS-tagged protein was purified by affinity chromatography and gel filtration using standard protocols[23], concentrated to 250 μM and buffer exchanged into 20 mM Sodium Phosphate (pH 6.7), 50 mM NaCl, 1 mM DTT. The protein backbone resonances were assigned after analysis of HNCA, HNCOCA, HNCACB, HNCO and 15N-NOESY-HSQC spectra according to standard protocols[57]. Peptide-bound ggSOCS1$^{ΔSOCSbox}$ spectra were assigned via the same experiments after addition of 400μM JAK1 activation loop peptide. Analysis of ggSOCS1$^{Δ-SOCSbox}$ binding to JAK2$^{KD}$ +/− JAK1 activation loop peptide were performed by collecting $^1$H–$^{15}$N SOFAST HMQC[58] spectra of 50 μM ggSOCS1$^{ΔSOCSbox}$ in the presence of a 2-fold excess of JAK2$^{KD}$ and/or JAK1 activation loop peptide. NMR analysis of ggSOCS1$^{ΔSOCSboxΔKIR}$ (residues 57–166) was performed via the same methodology.

**Isothermal titration calorimetry**. Isothermal calorimetric titrations were performed with a Microcal ITC200 (GE Healthcare). SOCS1/ElonginBC was purified by size exclusion chromatography and the running buffer (20 mM Tris-HCl pH 7.5, 150 mM NaCl, 2 mM β-mercaptoethanol) used to dissolve the phosphopeptides. Experiments were performed at 25 °C. Solutions of 10–40 μM SOCS1/ElonginBC in the cell were titrated by injection of a total of 39 μL of 80–400 μM of peptide. Two initial injections of 0.4 μL were performed and removed from the analysis. The heat of dilution of the peptide into buffer was determined and subtracted from the raw data of the binding experiment. The data were analysed using the evaluation software Origin 7.0.

**Analytical gel filtration**. SOCS1/elongin BC, SOCS1$^{ΔKIR}$ /elongin BC and/or JAK2 kinase domain were mixed together and diluted to a final volume of 500 μL in GF running buffer (10 mM Tris pH 7.5, 150 mM NaCl, 2 mM TCEP). Samples were loaded onto a S200 or S75 10/300 size exclusion chromatography column (GE Lifesciences) which had been pre-equilibrated in GF running buffer and A$_{280}$ was plotted as a function of elution volume.

**Data availability**. Atomic coordinates for the crystal structures have been deposited in the Protein Data Bank under accession numbers 6C7Y (SOCS1/JAK1) and 6C5X (SOCS1/elonginB/elonginC). Other data are available from the corresponding author upon request.

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

## Acknowledgements

This work was supported by the Cancer Council Victoria (Grant-in-aid 1065180) and the National Health and Medical Research Council (NHMRC) Australia (Project grant #1122999, Program grant #1113577), an NHMRC IRIISS grant 9000220, and a Victorian State Government Operational Infrastructure Scheme grant. J.J.B., J.M.M. and N.A.N. are supported by NHMRC fellowships and NPDL by an Australian Postgraduate Award. We thank the scientists at the MX beamline at the Australian Synchrotron. Crystallization trials were performed at CSIRO collaborative crystallization centre (C3).

## Author contributions

N.J.K. and N.P.D.L. carried out the crystallographic data collection, structure determination and refinement. J.J.B., A.L. and N.P.D.L. performed inhibition and biochemical assays J.M.M., A.L., N.P.D.L., E.W., I.S.L. and J.J.B. performed protein expression and purification experiments. S.Y., K.C. and J.J.B. performed NMR analyses. All authors commented on the manuscript. J.J.B., N.J.K., N.P.D.L. and N.A.N. designed the experiments, analyzed the data, supervised the project and wrote the paper.
