## [Peer Review File · Nature Communications]

Reviewers' Comments:

Reviewer #1 (Remarks to the Author):

The authors describe structural and functional studies of SOCS1 that reveal its mechanism of inhibition of JAK family kinases, as well as its assembly with the Elongin B/C complex. ALthough related SOCS proteins have previously been similarly characterized (in particular SOCS3), SOCS1 has distinct functional properties and has previously been resistant to structural studies. The crystal structure of a JAK1/SOCS1 complex described here provides for the first time a clear structural understanding of how the kinase inhibitory region (KIR) of SOCS1 inhibits JAKs by acting as a pseudo-substrate inhibitor at the peptide substrate binding site. The SOCS1/elonginB/C crystal structure confirms its expected similarity with previously described complexes with other SOCS family members. Extensive functional studies reveal the relevance of each residue of the KIR for mediating inhibition, and of an additional contact between Jak1 and the SOCS1 SH2 domain. Careful NMR-based structure/function studies also reveal that although the SOCS1 SH2 domain can bind peptides derived from the phosphorylated JAK2 activation loop, it does not engage the phosphorylated activation loop when bound to the intact Jak2 kinase domain, raising interesting questions about the role of the SH2 domain in directing the specificity of SOCS1 and other family members.

This work is an important advance in our understanding of SOCS1, and it also provides a model for understanding the KIR region of other family members. The crystallographic work appears to have been carefully executed and the structures well-refined. The manuscript is suitable for publication in its current form, and it should receive a high priority for publication in Nature Communications. A few minor comments:

1. A few more words about how this structure compares to the SOCS3/JAK structure would be helpful - in particular with respect to the KIR region - was this not included in prior structure, or was it disordered or differently ordered? Are the SH2 domain contacts with the kinase essentially the same?
2. The word "exclusively" is distracting/misleading in the describing the interaction of the KIR with JAK1. Could be interpreted to refer to the SOCS/JAK interaction as a whole, as odd to say that the interaction of the KIR is mediated exclusively by the KIR. Also, would start the description of the SOCS1/JAK interaction at a higher level, noting that it consists of contacts with both the SH2 domain and KIR region.
3. Might be worth a look to see if there are similarities worth mentioning between the KIR interaction seen here and the inhibitory interaction of MIG6 with EGFR. Also a pseudosubstrate inhibitory interaction that places a histidine in the phosphoacceptor position...(PDB 4ZJV, Park

et al. NSMB 2015)

4. The discussion regarding possible models for role of the SH2 domain feels a bit long/speculative, and I think distracts a bit from one of the major findings of this work - that binding to the phosphorylated Jak activation loop is probably not relevant. And doesn't offering the possibility of binding in trans in the context of a dimer conflict with the notion that it is sterically inaccessible in the context of the intact kinase domain?

-Michael Eck

Reviewer #2 (Remarks to the Author):

The manuscript by Liao et al. describes a novel structure of JAK1 bound to a highly relevant protein inhibitor SOCS1, as well as a second structure of SOCS1 bound to ElonginB/C. The authors go in to significant experimental detail to show validate the mechanism of JAK inhibition by SOCS1 that is suggested by their structure. This is the first example of a JAK bound to any sort of substrate, pseudo or otherwise, at sub 3Å resolution, which allows for detailed inspection of binding. The structure also builds significantly onto their previous work (published in NSMB in 2013) on the structure of JAK2/SOCS3/gp130 and elegantly describes why the two seem to have somewhat different JAK inhibition characteristics. The structural data collection and refinement and biochemical experiments appear to have been carried out in a skillful fashion, with a detailed methods section also presented. In my opinion, this is an important structure, and should be published in Nature Communications without significant delay. I do have a few comments below I would like the authors to clarify or add to the text.

- 1) I think the introduction would benefit from a sentence or two about the interactions between SOCS proteins and Elongins, to refresh the readers mind about this before the SOCS1/Elongin data is presented.
- 2) Have the authors mutated KIR His54 to Phe or Tyr to generate a more tightly bound SOCS1? Does a Tyr in this position get phosphorylated similarly to SOCS3? I may have missed it but this data (SOCS3 KIR Tyr mutant phosphorylation) should be mentioned in the text somewhere if possible.
- 3) Do the authors have data that shows SOCS1 binds phosphorylated JAK1, similar to SOCS1 and 3 binding to phosphorylated JAK2?
- 4) Could the authors speculate further in the discussion on why SOCS1 doesn't bind receptors well, compared to SOCS3? Any mutations that would change things? Is it the loop length in the SH2 specificity determining pocket (pY+3)? When they mention on p16 that these loops are the

shortest of any SH2 domain, is this SOCS SH2 domains, or all SH2 domains? Please clarify this last comment.

5) The authors claim that SOCS1 can inhibit JAK autophosphorylation, which I agree is very likely given the data presented. But on p15 they claim that SOCS3 "...only inhibits phosphorylation of downstream targets". I find this surprising given the similarities of the JAK1/SOCS1 and JAK2/SOCS3 structures, and the fact that SOCS1 and SOCS3 can bind both phosphorylated and non-phosphorylated JAK2. Could the authors please explain this comment in their response?

Reviewer #3 (Remarks to the Author):

Liau et al. present structural and biochemical data demonstrating how SOCS1 interacts with JAK tyrosine kinases to inhibit their catalytic activity. This study follows a previous structural study by these authors on a related SOCS protein, SOCS3, in complex with JAK2. In the current study, the authors determined a crystal structure of a complex between SOCS1 and JAK1, which was made possible by exploring numerous constructs of the ill-behaved SOCS1 protein. The structure shows that SOCS1, like SOCS3, inhibits JAKs (JAK1, JAK2, and TYK2, but not JAK3) by inserting an N-terminal segment (KIR) into the substrate binding groove of the kinase (as a pseudosubstrate), using an interaction between the SOCS1 SH2 domain and helix G of the kinase domain as an anchoring contact. Mutagenesis experiments confirm the importance of the KIR and the GQM motif in helix G (which is absent in JAK3) for the potent inhibition of JAK1 by SOCS1. The authors also report a crystal structure of SOCS1 in complex with Elongin BC, which explains why SOCS1 interacts relatively weakly with Cullin5, an E3 ligase scaffold protein.

Much of the rest of the study is focused on the binding target of the SOCS1 SH2 domain. Unlike the SH2 domain of SOCS3, which was shown to bind to phosphotyrosines on cytokine receptors such as gp130, the authors were not able, with significant effort expended, to identify a single plausible phosphorylation site on a cytokine receptor as the target for the SOCS1 SH2 domain. Interestingly, the phosphorylated activation loops of JAKs, presented as phosphopeptides, did bind with high affinity to the SOCS1 SH2 domain, but not the intact, phosphorylated activation loop (although there was a hint of binding when a type II JAK inhibitor was present). Based on their SOCS1-JAK1 structure, it is highly doubtful (from steric considerations, Fig. S5) that the SOCS1 KIR and SH2 domain can engage simultaneously the substrate binding groove and phosphorylated activation loop of the same kinase domain.

Despite the remaining conundrum regarding the target of the SOCS1 SH2 domain, this study

adds significantly to our understanding of the downregulation of JAKs by SOCS proteins and will be of great interest to those in the cytokine signaling field.

Response to Reviewers

We sincerely thank the reviewers for their kind comments and their excellent queries. Please see below for a detailed response.

Reviewer 1:

We sincerely thank the reviewer 1 for their comments and questions.

1. A few more words about how this structure compares to the SOCS3/JAK structure would be helpful - in particular with respect to the KIR region - was this not included in prior structure, or was it disordered or differently ordered? Are the SH2 domain contacts with the kinase essentially the same?

This is an excellent suggestion. We have added the following sentences to the results section (subheading “SOCS1 inhibits JAK by blocking substrate binding”):

“Both the KIR and SH2 domain driven interactions are conserved in the SOCS3/JAK2 structure with similar overall geometry.”

“As shown in Figure 2E, the majority of the individual contacts between the KIR and JAK are conserved in both SOCS1 and SOCS3. The significantly improved resolution of the SOCS1/JAK1 structure allows a detailed atomic-level analysis of this interaction (Supplementary Figure 1B) and shows that it is mediated by a continuous six residue segment of the KIR (His54 to Arg59), corresponding to residues -1 to +5 of this motif based on earlier definitions.”

2. The word "exclusively" is distracting/misleading in the describing the interaction of the KIR with JAK1. Could be interpreted to refer to the SOCS/JAK interaction as a whole, as odd to say that the interaction of the KIR is mediated exclusively by the KIR.

We have removed the word “exclusively”. We were attempting to explain that its only the first half of the KIR that is involved in the interaction (and not the second half) however we agree it is confusing.

Also, would start the description of the SOCS1/JAK interaction at a higher level, noting that it consists of contacts with both the SH2 domain and KIR region.

We have added the following sentence to the very beginning of our description of the structure:

“As shown in Figure 2A, SOCS1 binds to JAK using both its SH2 domain and kinase inhibitory region. The interface between the two proteins is 1159 Å² and the KIR of SOCS1 is responsible for approximately half of this buried surface (Figure 2B).”

3. Might be worth a look to see if there are similarities worth mentioning between the KIR interaction seen here and the inhibitory interaction of MIG6 with EGFR. Also a pseudosubstrate inhibitory interaction that places a histidine in the phosphoacceptor position...(PDB 4ZJV, Park et al. NSMB 2015)

We sincerely thank the reviewer for this insight as we were unaware of that structure. Indeed both proteins use a histidine to mimic substrate and both adjacent residues are also similar, despite the overall conformation of both motifs being different. We have now included several sentences on this in the discussion:

“Another kinase-inhibitory protein, Mig6 (an EGFR inhibitor) acts as a substrate competitor and also places a histidine sidechain in the substrate binding site³⁵. The kinase inhibitory segment of Mig6 adopts a beta-hairpin configuration and is anchored to its target kinase by a distal element upstream of it. The KIR of SOCS1 is also anchored to its target by regions of the protein distal to in it in the primary sequence and it seems likely that both proteins require a higher affinity interaction with their targets than can be provided by the kinase inhibitory segments alone.”

4. The discussion regarding possible models for role of the SH2 domain feels a bit long/speculative, and I think distracts a bit from one of the major findings of this work - that binding to the phosphorylated Jak activation loop is probably not relevant. And doesn't offering the possibility of binding in trans in the context of a dimer conflict with the notion that it is sterically inaccessible in the context of the intact kinase domain?

We have been very careful not to rule out an interaction between the SH2 domain of SOCS1 and the activation loop of JAK *in vivo*. To us it seems an odd coincidence that the four highest-affinity phosphopeptides we found (out of >35 tested) are derived from the activation loops if SOCS1 were to never bind them as part of its mechanism of action. However we tested for such an interaction exhaustively (described in figure 7 and also in many experiments not shown here) and could not detect one except under the strange circumstances of when JAK1 was bound by a type II inhibitor. Type II inhibitors are known to cause disorder in the activation loops of many kinases (including JAK) and we feel that particular result leaves open the possibility of some type of interaction inside the cell acting similarly and disordering the loop. Nevertheless we agree with the reviewer that we spent too long discussing this in the manuscript and have now shortened that section significantly (the penultimate paragraph of the discussion).

Reviewer 2

We sincerely thank the reviewer 1 for their comments and questions.

1) I think the introduction would benefit from a sentence or two about the interactions between SOCS proteins and Elongins, to refresh the readers mind about this before the SOCS1/Elongin data is presented.

This is an excellent suggestion and we have added the following to the introduction:

“The SOCS box of all SOCS proteins are found associated with an adapter complex, elonginBC. This association allows recruitment of an E3 ubiquitin ligase scaffold (Cullin5) to catalyse the ubiquitination of signalling intermediates recruited by their SH2 domains”

2) Have the authors mutated KIR His54 to Phe or Tyr to generate a more tightly bound SOCS1? Does a Tyr in this position get phosphorylated similarly to SOCS3? I may have missed it but this data (SOCS3 KIR Tyr mutant phosphorylation) should be mentioned in the text somewhere if possible.

We have now made the H54Y mutant and shown that it is efficiently phosphorylated by JAK1 although it does not alter the affinity significantly (in fact, much as the H54A mutation it makes it slightly weaker). We have now described this in the results:

“In SOCS3, when this residue is mutated to tyrosine it is efficiently phosphorylated by JAK and the same is true for SOCS1 (Supplementary Figure 1C).”

And we have added an extra panel to Supplementary Figure 1 and placed the mutant IC₅₀ data into Figure 3.

3) Do the authors have data that shows SOCS1 binds phosphorylated JAK1, similar to SOCS1 and 3 binding to phosphorylated JAK2?

We have had a lot of trouble performing such experiments because we cannot isolate quantitatively phosphorylated JAK1, unlike JAK2. Our best attempts have led to ~20% phosphorylated JAK1. Nevertheless it is clear that SOCS1 does bind to phosphorylated JAK because it is able to inhibit it—unphosphorylated JAK1 is inactive (in our hands) and therefore any inhibition we see is due to SOCS1 interacting with the phosphorylated form of the kinase. And of course SOCS1 is able to bind to phosphorylated JAK2 as the reviewer notes.

4) Could the authors speculate further in the discussion on why SOCS1 doesn't bind receptors well, compared to SOCS3? Any mutations that would change things? Is it the loop length in the SH2 specificity determining pocket (pY+3)? When they mention on p16 that these loops are the shortest of any SH2 domain, is this SOCS SH2 domains, or all SH2 domains? Please clarify this last comment.

This is an excellent question. There doesn't seem to be anything in particular stopping SOCS1 from binding to receptor motifs. We think it is just that the system has not evolved to require it. For example there are no conserved tyrosines on the interferon alpha or gamma receptors that SOCS1 can bind to. SOCS1 has a much wider range of target cytokines than SOCS3 which is specific for only a few. It may be that bypassing receptor involvement allows SOCS1 to target more cytokines. Another issue is why SOCS1 can bind to the activation loop peptides with high affinity, answering that question will require a structure of SOCS1 bound to such a peptide and we have been unable to crystallise such a complex. Our comments about the short loops that form the pY+3 pocket are not intended to imply that SOCS1 is incapable of binding to a receptor motif but rather that it would make it possible for it to bind to the JAK activation loop if there was something that promoted the disordering of that loop. The two loops in SOCS1 are indeed the shortest of ANY SH2 domain (as far as we can tell), not just of the SOCS SH2 domains. We have tried to make these thoughts clear by altering the discussion (whilst avoiding being too speculative). We have inserted the following sentences:

“Therefore, although SOCS1 can bind to certain receptor motifs *in vitro*, data shown here and by others suggest SOCS1 does not require receptor-binding *in vivo*.”

“It is notable that, to our knowledge, the two loops within the SOCS1 SH2 domain that form the pY+3 pocket (the specificity-determining pocket within most SH2 domains) are the shortest of any SH2 domain in the human proteome”

5) The authors claim that SOCS1 can inhibit JAK autophosphorylation, which I agree is very likely given the data presented. But on p15 they claim that SOCS3 "...only inhibits phosphorylation of downstream targets". I find this surprising given the similarities of the JAK1/SOCS1 and JAK2/SOCS3 structures, and the fact that SOCS1 and SOCS3 can bind both

phosphorylated and non-phosphorylated JAK2. Could the authors please explain this comment in their response?

This is an excellent question. The data showing that SOCS3 does not inhibit JAK autophosphorylation *in vivo* but that SOCS1 does was published first by Yoshimura (reference 4) and we have also observed such a phenomenon previously (Linossi et al., PLOS one 2013). Our best explanation is that due to the enhanced affinity of SOCS1 for its target JAK (compared with SOCS3) it is able to target the kinase in an unphosphorylated state efficiently and prevent it from being phosphorylated. SOCS3 on the other hand, which binds to JAK with much lower affinity cannot do so. SOCS3 seems designed to inhibit JAK only with the help of an associated cytokine receptor, which contain a SOCS3 binding site, in order to anchor it nearby. This is presumably why SOCS3 is more specific than SOCS1. Receptors are only phosphorylated after JAK itself becomes activated and therefore SOCS3 has to wait for JAK to become activated (or re-activated after a phosphatase-mediated dephosphorylation event) in order to inhibit it. By bypassing the need for receptor-binding SOCS1 can inhibit JAK from becoming phosphorylated itself. Although this is rather speculative we have added a short sentence in the discussion along those lines.

“The ability of SOCS1 to bind with such high affinity to dephosphorylated JAK, without the aid of a phosphorylated cytokine receptor may explain why, *in vivo* SOCS1 can inhibit JAK autophosphorylation (as well as downstream phosphorylation) in contrast to SOCS3 which cannot.”

Reviewer 3

We sincerely thank the reviewer for their kind comments regarding our manuscript.